# The Impact of *Nigella sativa* Essential Oil on T Cells in Women with Hashimoto’s Thyroiditis

**DOI:** 10.3390/antiox12061246

**Published:** 2023-06-09

**Authors:** Klaudia Ciesielska-Figlon, Karolina Wojciechowicz, Agnieszka Daca, Adam Kokotkiewicz, Maria Łuczkiewicz, Jacek Maciej Witkowski, Katarzyna Aleksandra Lisowska

**Affiliations:** 1Department of Physiopathology, Faculty of Medicine, Medical University of Gdańsk, 80-210 Gdańsk, Poland; 2Division of Pathology and Experimental Rheumatology, Faculty of Medicine, Medical University of Gdańsk, 80-210 Gdańsk, Poland; 3Department of Pharmacognosy, Faculty of Pharmacy, Medical University of Gdańsk, 80-210 Gdańsk, Poland

**Keywords:** *Nigella sativa*, essential oil, Hashimoto’s thyroiditis, apoptosis, proliferation, immunomodulation

## Abstract

Background: Hashimoto’s thyroiditis (HT) is an autoimmune disease mediated by T cells. It is characterized by the presence of thyroid autoantibodies in the serum, such as anti-thyroid peroxidase antibodies (TPO-Ab) and anti-thyroglobulin antibodies (TG-Ab). The essential oil extracted from *Nigella sativa* seeds is rich in bioactive substances, such as thymoquinone and cymene. Methods: Therefore, we examined the effect of essential oil from *Nigella sativa* (NSEO) on T cells from HT patients, especially their proliferation capacity, ability to produce cytokines, and susceptibility to apoptosis. Results: The lowest ethanol (EtOH) dilution (1:10) of NSEO significantly inhibited the proliferation of CD4^+^ and CD8^+^ T cells from HT patients and healthy women by affecting the percentage of dividing cells and the number of cell divisions. In addition, 1:10 and 1:50 NSEO dilutions induced cell death. Different dilutions of NSEO also reduced the concentration of IL-17A and IL-10. In healthy women, the level of IL-4 and IL-2 significantly increased in the presence of 1:10 and 1:50 NSEO dilutions. NSEO did not influence the concentration of IL-6 and IFN-γ. Conclusions: Our study demonstrates that NSEO has a strong immunomodulatory effect on the lymphocytes of HT patients.

## 1. Introduction

Hashimoto’s thyroiditis (HT), also known as chronic autoimmune thyroiditis, is an autoimmune disease characterized by lymphocytic infiltration into and gradual damage to the thyroid parenchyma and the production of organ-specific anti-thyroid antibodies: anti-thyroid peroxidase (TPO-Ab) and anti-thyroglobulin (TG-Ab) [1,2,3]. Studies using reverse polymerase chain reaction to assess cytokine gene expression within the thyroid gland showed increased mRNA levels of IL-2, IL-4, IL-10, and INF-γ [4]. As with other diseases of autoimmune etiology, Hashimoto’s thyroiditis is caused by a combination of genetic and environmental factors, leading to a loss of immune tolerance and resulting in the destruction of thyroid tissue [5]. Based on epidemiological studies, HT most commonly affects women between 30 and 50 years [1].

The etiology of Hashimoto’s thyroiditis is complex. A critical moment in the pathogenesis of HT is the formation of autoreactive cells directed against the thyroid gland, which may result from a loss of immunological tolerance to its tissues. Immune tolerance is developed during the perinatal period. At that time, immature lymphocytes are exposed to autoantigens, and self-tolerance to autoantigens is conditioned by the clonal deletion and induced anergy of autoreactive T cells [6]. However, these mechanisms may fail, and some cells recognizing autoantigens enter the bloodstream, which leads to a breakdown of auto-tolerance, in this case, against thyroid antigens. This process may be caused by overexposure to thyroid antigens and exposure to environmental antigens similar to autoantigens [7]. The central role in this process is attributed to T helper 1 (Th1) cells, a CD4^+^ effector T cell lineage. Th1 cells are responsible for the lymphocytic inflammatory infiltration into the thyroid gland, which at a later stage causes thyroiditis and damage to the thyroid gland. They produce IFN-γ and activate the infiltration of macrophages into the thyroid gland, thereby inducing the secretion of various cytokines and the progressive damage of thyrocytes by apoptosis. Activated cytotoxic lymphocytes and macrophages are directly responsible for destroying thyroid follicular cells [8]. Furthermore, through stimulation with cytokines, mainly interleukin-1 (IL-1) secreted by antigen-presenting cells (APCs) and Th1 cells, follicular thyroid cells are induced to express molecules involved in the regulation of programmed cell death, including Fas and FasL. The result of this stimulation can be cell apoptosis [9].

In addition, CD8^+^ (cytotoxic) T cells (CTLs) may cause thyrocyte destruction through the Fas–FasL pathway [9]. CTLs produce cytotoxic granules, such as perforin and granzymes (including granzyme B) [10]. The perforin molecule forms a pore in the cellular membrane of target cells, and granzyme B activates proapoptotic molecules, such as caspases and cytochrome c [11]. CTLs detected in HT patients could be one of the causative factors for thyrocyte destruction and hypothyroidism [12]. TPO- and Tg-specific CD8^+^ T cells are present in the peripheral blood and thyroid gland of HT patients, where they induce the lysis of target cells [13].

In addition to Th1 cells, Th2 cells are also involved in the pathogenesis of HT. They stimulate B cells to produce antibodies against thyroid antigens, such as TPO and TG, thereby leading to thyroid tissue inflammation [14]. Furthermore, Th2 cells produce cytokines, such as IL-4, IL-5, IL-6, and IL-10, further stimulating the production of anti-thyroid antibodies. Hence, thyroid cells have been shown to produce cytokines, such as IL-1, IL-6, IL-8, TNF-α, and TGF-β, leading to their destruction [15].

There is increasing interest worldwide in alternative therapies based on herbal medicine or herbal extracts and their supplementation in chronic diseases, including Hashimoto’s thyroiditis. Among the medicinal herbs gaining more and more attention is *Nigella sativa* (NS), also known as black cumin or black seed, which belongs to the Ranunculaceae family, cultivated in many regions, such as the central and eastern Mediterranean, southern Europe, India, Pakistan, Syria, and Turkey [16]. Black seeds are a source of active ingredients characterized by antioxidant, anti-inflammatory, and immunomodulatory properties [17]. NS seed supplementation has been shown to improve the efficacy of specific immunotherapy in patients with allergic rhinitis [18], rheumatoid arthritis (RA) [19], or asthma [20]. In women with RA, treatment with NS seed oil led to a significant reduction in serum C-reactive protein (CRP) levels and an improvement in the number of swollen joints compared to the control group [19]. In addition, a reduction in the percentage of CD8^+^ (cytotoxic) T cells and an increase in CD4^+^CD25^+^ cells were also observed compared to the placebo. In young patients with asthma, supplementation with NS seed oil caused a significant reduction in Th17 cells and increased regulatory T cells (Tregs) [20].

The presented studies demonstrate the strong therapeutic potential and use of NS in treating autoimmune diseases. In hypothyroidism, the therapeutic effect of NS is attributed to its antioxidant action [21,22]. In a study by Farhangi et al. [23], an 8-week powdered NS treatment caused a decrease in TSH and TPO-Ab and increased serum triiodothyronine (T3) levels in HT patients. Similar results were obtained by Tajmiria et al. [24], who also investigated the effect of NS powder supplementation on thyroid function. Additionally, the authors demonstrated a reduction in serum IL-23, an inflammatory cytokine involved in amplifying and stabilizing Th17 cells. NS seed oil was also shown to have antioxidant activity that may reduce propylthiouracil (PTU)-induced oxidative stress and damage to thyroid follicles, which may have important therapeutic implications in hypothyroidism [25].

The reduction in serum TPO-Ab levels after treatment with NS seed oil or powder in different studies can be explained by its immunomodulatory effects, which have been previously confirmed by its protective role against several autoimmune diseases, including type 1 diabetes and experimental autoimmune encephalomyelitis (EAE) [26,27]. Furthermore, Avci et al. [28] demonstrated that it is the essential oil (EO) obtained from NS (NSEO) that is responsible for increasing the concentration of total triiodothyronine (tT3) in hypothyroid and hyperthyroid models in rats. Still, there are not enough studies verifying the therapeutic effect of Nigella sativa in patients suffering from autoimmune diseases. Therefore, in order to explain the potential immunomodulatory properties of *Nigella sativa*, this study investigates the influence of NSEO on activated T cells, including their proliferation capacity, ability to produce cytokines, and susceptibility to apoptosis, in women with Hashimoto’s thyroiditis.

## 2. Materials and Methods

### 2.1. Study Group

The study group consisted of 9 women diagnosed with Hashimoto’s thyroiditis, with a mean age of 39 ± 8 years. The control group consisted of 9 healthy women with a mean age of 36 ± 8 years. The control group did not take any medication that could influence the immune system. All participants were informed about the purpose of the study and provided their written informed consent to participate in the study. The Independent Bioethics Committee for Scientific Research approved the study (consent no. NKBBN/417-233/2019, received on 10 April 2019). All experiments were conducted following relevant guidelines and regulations.

### 2.2. Plant Material and Essential Oil Isolation

The seeds of black cumin were obtained from Makar Bakalie, Poland. The material was stored in the dark in a hermetically sealed container and processed with convection drying (105 °C for 4 h, Binder FD oven, Tuttlingen, Germany). For NSEO isolation, a 100 g sample of NS seeds was frozen in liquid nitrogen. Next, the still-frozen material was moved into a round-bottom flask, water with an antifoam agent (Sigma Aldrich Inc., Saint Louis, MO, USA) was added, and the flask was connected to an etheric oil distillation apparatus (Carl-Roth, Karlsruhe, Germany). NSEO isolation and GC analysis were performed in the Department of Pharmacognosy of the Medical University of Gdańsk and the Department of Analytical Chemistry of the Gdańsk University of Technology. The details of the processes and parameters are described in a previous paper [29].

### 2.3. PBMC Isolation and Stimulation

About 25 mL of peripheral venous blood was collected from each subject into EDTA blood collection tubes. Peripheral blood mononuclear cells (PBMCs) were next isolated by centrifuging the blood in a gradient on Histopaque^®^-1077 (Sigma Aldrich Inc., Saint Louis, MO, USA). Next, PBMCs were stained with Violet Proliferation Dye 450 (VPD450; Becton Dickinson, Franklin Lakes, NJ, USA) for 12 min in the dark at 37 °C according to the manufacturer’s protocol. Afterward, the cells were washed in phosphate-buffered saline (PBS) (EURx, Gdańsk, Poland), after which they were resuspended in complete culture medium (RPMI 1640 supplemented with 10% fetal bovine serum, 2 mM L-glutamine, 100 U/mL of penicillin, and 100 μg/mL of streptomycin) at a concentration of 1.5 million cells per 1.5 medium.

The cells were then incubated with an immobilized (tissue culture plate-bound) anti-CD3 monoclonal antibody (BD Pharmingen, San Diego, CA, USA) for 24 h to activate the lymphocytes. Four (1:10, 1:50, 1:100, 1:500, and 1:1000) ethanol (EtOH) dilutions of NSEO were prepared. After 24 h, 3.75 µL of each EtOH NSEO dilution was added to preactivated cells. The final NSEO concentrations in the cell culture were 1:10–0.025%, 1:50–0.005%, 1:100–0.0025%, 1:500–0.0005%, and 1:1000–0.00025%. The controls consisted of cells stimulated with an anti-CD3 antibody (in experiments assessing proliferation, apoptosis, and necrosis) and unstimulated (US) cells (in experiments assessing cytokine production).

Stimulated cells were harvested after 72 h and 120 h. They were then stained with the following antibodies conjugated with fluorescent dyes: PerCP-conjugated anti-CD4 and APC-H7-conjugated anti-CD8 (BD Pharmingen, San Diego, CA, USA). In addition, cells were also stained with PE-conjugated annexin V and 7-aminoactinomycin D (7-AAD) according to the manufacturer’s protocol (BD Pharmingen, San Diego, CA, USA) and analyzed with flow cytometry using a FACSVerse instrument (Becton Dickinson, Franklin Lakes, NJ, USA).

### 2.4. Measurement of Cytokines in Cell Culture Supernatants

We performed quantitative cytometric fluorescence analysis with the FACSAria III cytometer (Becton Dickinson, Franklin Lakes, NJ, USA). The BD™ Cytometric Bead Array (CBA) Human Th1/Th2/Th17 Cytokines Kit (BD Biosciences, San Jose, CA, USA) was used according to the manufacturer’s protocol to determine the level of seven different cytokines in the cell culture supernatant samples from HT patients and healthy controls: IL-2, IL-4, IL-6, IL-10, TNF, IFN-γ, IL-17A. The kit performance was optimized to analyze physiologically relevant concentrations (pg/mL levels) of specific cytokine proteins in tissue culture supernatants and serum samples. The limit of detection for IL-2 was 2.6 pg/mL; IL-4, 4.9 pg/mL; IL-6, 2.4 pg/mL; IL-10, 4.5 pg/mL; TNF, 3.8 pg/mL; IFN-γ, 3.7 pg/mL; and IL-17A, 18.9 pg/mL.

### 2.5. Analysis and Statistics

Lymphocytes were sorted based on forward scatter (FSC) and side scatter (SSC) characteristics and their positivity for surface antigens (CD4, CD8), as previously described [29]. Cytometric analysis was performed using FCSalyzer software (copyright © 2012–2019 Sven Mostböck).

The dividing cell tracking (DCT) method was applied to examine the cell proliferation kinetics [30]. It uses VPD450, which becomes fluorescent and covalently binds to proteins within the cells after cleavage by esterase within viable cells. As viable cells divide, the VPD450 dye is distributed uniformly between daughter cells, so each daughter cell retains approximately half of the VPD450 fluorescence intensity of its parent cell. In Figure 1, non-dividing cells are indicated with marker 1 (M1), while proliferating cells are indicated with M2. The number of divisions, on the contrary, is shown as the number of division peaks on the histogram within cells marked with M2. Annexin V and 7-AAD staining was used to identify cells as alive (cells negative for both annexin V and 7-AAD), in early apoptosis (annexin V-positive 7-AAD-negative cells), in late apoptosis (cells positive for both annexin V and 7-AAD), and necrotic (cells only 7-AAD-positive).

Data, including the standard curve range for a given cytokine (IL-2, IL-4, IL-6, IL-10, TNF, IFN-γ, or IL-17A), were statistically analyzed using GraphPad Prism software, version 9 (GraphPad Software, San Diego, CA, USA). Significance tests were selected according to the data distribution, with a significance level of *p* < 0.05.

## 3. Results

### 3.1. Influence of NSEO on T Cell Proliferation in HT Patients and Healthy Women

Figure 2 presents the number of cell divisions of lymphocytes stimulated with immobilized monoclonal anti-CD3 antibody in the presence of different EtOH dilutions of NSEO. In healthy women, a decrease in the number of cell divisions of CD4^+^ (Figure 2a) and CD8^+^ (Figure 2c) cells after 72 h of stimulation was observed in the presence of the lowest (1:10) dilution of NSEO compared to cells stimulated with anti-CD3 antibody alone. In HT patients, there was also a decrease in the number of cell divisions of CD4^+^ and CD8^+^ cells in the presence of 1:10 NSEO dilution after 72 h of stimulation. Additionally, there was a decrease in the number of cell divisions of CD8^+^ cells in the presence of 1:50 and 1:100 NSEO in HT patients after 72 h (Figure 2c). After 120 h, there was also a decrease in the number of cell divisions of CD4^+^ (Figure 2b) and CD8^+^ (Figure 2d) cells from HT patients and healthy women when cells were incubated with 1:10 NSEO dilution. Significant differences were also seen between higher NSEO dilutions (1:500 and 1:1000) and 1:10 NSEO dilution, especially in healthy women.

Figure 3 shows the changes in the percentage of dividing CD4^+^ and CD8^+^ cells. After 72 h, we observed a decrease in the percentage of proliferating cells in the presence of 1:10 NSEO in healthy women and HT patients for each T cell subpopulation (CD4^+^ and CD8^+^); see Figure 3a,c. In HT patients, the decrease in the percentage of proliferating CD4+ cells was significant compared to cells incubated with anti-CD3 antibody alone (Figure 3a). Meanwhile, in healthy women, the decrease in the percentage of proliferating CD8^+^ cells was significant compared to cells incubated with anti-CD3 antibody alone and cells incubated in the presence of 1:500 and 1:1000 NSEO dilutions (Figure 3c). After 120 h, there was also a decrease in the percentage of proliferating CD4^+^ and CD8^+^ cells in healthy women and HT patients when cells were incubated with 1:10 NSEO dilution compared to cells incubated with anti-CD3 antibody alone (Figure 3b,d). In healthy women, the decrease was also significant when compared to cells incubated with 1:100 or 1:500 NSEO dilution. In HT patients, the decrease was significant compared to cells incubated with 1:500 NSEO dilution but not when cells were incubated with 1:100 NSEO dilution.

### 3.2. Influence of NSEO on Lymphocyte Apoptosis in HT Patients and Healthy Women

The most significant differences in the percentages of live, apoptotic, or necrotic cells were seen between lymphocytes stimulated only with anti-CD3 antibody and cells incubated in the presence of 1:10 and 1:50 dilutions of NSEO. After 72 h, in healthy women and HT patients, there was a significant decrease in the percentage of living cells after incubation with 1:10 and 1:50 NSEO (Figure 4a), with a simultaneous increase in cells in late apoptosis (Figure 4c) when compared with cells incubated with anti-CD3 antibody alone or in combination with higher dilutions (1:100, 1:500, and 1:1000). The percentage of cells in early apoptosis decreased after 72 h incubation with 1:10 dilution of NSEO in healthy women and HT patients (Figure 4b). In healthy people, there was no change in the percentage of necrotic cells after NSEO treatment (Figure 4d). In HT patients, there was an increase in necrotic cells after incubation with 1:10 and 1:50 NSEO dilutions compared to cells incubated with anti-CD3 antibody alone or in combination with higher dilutions of NSEO.

After 120 h, there was a decrease in the percentage of living cells after incubation with the lowest NSEO dilution (1:10) compared to other stimulation variants in healthy women and HT patients. There were significant differences between the lowest dilutions (1:10 and 1:50) in healthy controls and between 1:10 and higher dilutions, such as 1:100, 1:500, and 1:1000, in HT patients (Figure 4e). The percentages of cells in late apoptosis (Figure 4g) and necrosis (Figure 4h) were higher in the presence of 1:10 and 1:50 NSEO dilutions in healthy women and in the presence of 1:10 NSEO in HT patients. There was a considerable statistical difference in those variants, especially between the control variant and a low dilution of NSEO and between the 1:10 variant and higher dilutions, such as 1:100, 1:500, and 1:1000. The changes in the percentage of cells in early apoptosis were similar to those observed after 72 h. There was a significant difference between the control variant and 1:10 NSEO in healthy controls and HT patients. There was also a significant difference between the 1:10 and 1:100 variants in healthy controls and 1:500 dilution in HT patients (Figure 4f).

### 3.3. Influence of NSEO on the Production of Cytokines in HT Patients and Healthy Women

We also compared the concentrations of cytokines (IL-2, IL-4, IL-6, IL-10, TNF, IFN-γ, and IL-17A) in cell culture supernatants after 72 h of stimulation with immobilized monoclonal anti-CD3 antibody in the presence of different EtOH dilutions of NSEO. In addition, supernatants from wells where cells were incubated without stimulation served as an unstimulated (US) control.

In both healthy women and HT patients, IL-2 decreased after stimulation with anti-CD3 antibody alone (Figure 5a). In healthy women, the level of IL-2 significantly increased in the presence of 1:10 and 1:50 NSEO dilutions and decreased in the presence of 1:100, 1:500, and 1:1000 NSEO dilutions. Meanwhile, in HT patients, the level of IL-2 significantly decreased in the presence of 1:50, 1:100, and 1:1000 NSEO dilutions.

In healthy women, the IFN-γ level increased after stimulation with anti-CD3 antibody alone or in the presence of 1:10, 1:100, and 1:500 NSEO dilutions compared to the unstimulated control (Figure 5b). In HT patients, the IFN-γ level also increased after stimulation with anti-CD3 antibody alone compared to the unstimulated control. In addition, in the presence of 1:10, 1:500, and 1:1000 NSEO dilutions, the IFN-γ level also increased compared to the unstimulated control, but it was visibly lower compared to anti-CD3 antibody alone.

In addition, in both study groups, the TNF level increased after stimulation with anti-CD3 antibody compared to the unstimulated control (Figure 5c). In healthy women, the level of TNF significantly increased in the presence of 1:10, 1:50, and 1:100 NSEO dilutions compared to the unstimulated control. Meanwhile, in HT patients, the level of TNF significantly decreased in the presence of 1:10 and 1:100 NSEO dilutions compared to anti-CD3 antibody stimulation.

In healthy women and HT patients, there was an increase in IL-17A levels after stimulation with anti-CD3 antibody compared to the unstimulated control (Figure 5d). In healthy women, IL-17A levels decreased in the presence of 1:10 and 1:50 NSEO dilutions compared to anti-CD3 antibody. In addition, IL-17A levels increased in the presence of 1:100, 1:500, and 1:1000 NSEO compared to the unstimulated control, but the levels were visibly lower compared to anti-CD3 antibody alone. In HT patients, IL-17A levels decreased in the presence of 1:10 NSEO, but in the presence of 1:500 NSEO, IL-17A levels increased compared to the unstimulated control.

The level of IL-4 in supernatants after stimulation with anti-CD3 antibody alone did not change significantly compared to the unstimulated control in HT patients and healthy women (Figure 6a). However, the IL-4 level increased in the presence of 1:10 and 1:50 NSEO in healthy women and in the presence of 1:10 NSEO in HT patients. However, the level of IL-4 in the presence of 1:10 NSEO increased to a lesser extent in HT patients compared to healthy women (*p* = 0.0002, Mann–Whitney U test). In healthy women, there were differences between the unstimulated control and cells stimulated with 1:10 and 1:50 NSEO dilutions; between cells stimulated only with anti-CD3 antibody and with 1:10 and 1:50 NSEO dilutions; between cells stimulated with 1:10, 1:100, 1:500, and 1:1000 NSEO dilutions; and between cells stimulated with 1:50, 1:500, and 1:1000 NSEO dilutions. In HT patients, the difference was only between the unstimulated control and cells stimulated with 1:10 NSEO dilution.

There was an increase in IL-6 levels after stimulation with anti-CD3 antibody compared to the unstimulated control in both study groups (Figure 6b). In healthy women, the level of IL-6 also increased in the presence of 1:10, 1:50, 1:100, and 1:500 NSEO dilutions compared to the unstimulated control. In HT patients, IL-6 levels increased in the presence of 1:10, 1:50, 1:500, and 1:1000 NSEO dilutions compared to the unstimulated control.

There was an increase in the IL-10 level after stimulation with anti-CD3 antibody compared to the unstimulated control in both study groups (Figure 6c). In healthy women, the IL-10 level decreased in the presence of 1:10 and 1:50 NSEO dilutions compared to anti-CD3 antibody alone. Compared to the unstimulated control, the IL-10 level increased in the presence of 1:500 and 1:1000 NSEO dilutions. In HT patients, the level of IL-10 was significantly lower in unstimulated cells than in those exposed to 1:500 NSEO dilution.

## 4. Discussion

The pathogenesis of HT is determined by a disturbed immune system, manifested by the accumulation of autoreactive lymphocytes and macrophages with the simultaneous loss of immune tolerance to one’s own tissues, which in turn leads to the destruction of the thyroid gland and the development of hypothyroidism [2]. CD4^+^ T cells play a significant role in HT immunopathogenesis. When they get activated, they differentiate into different subtypes, depending on the cytokines released. By secreting interferon-gamma (IFN-γ), Th1 cells control macrophage-dependent cell-mediated immunity. Meanwhile, Th2 cells secrete IL-4, IL-5, or IL-10 and regulate B cell responses [31]. IL-12 is crucial for activating Th1 responses, while IL-6 promotes the IL-4-dependent induction of Th2 differentiation. In addition, CD8^+^ T cells may cause thyrocyte destruction through the Fas–FasL pathway [9] or perforin and granzymes [10]. Suppression of T cell responses could play a clinical role in the development and course of Hashimoto’s thyroiditis, even at the stage of full progression of the disease, because the thyroid tissue of HT patients is infiltrated by CD69^+^ and CD25^+^, with moderate numbers of Foxp3+ cells. In addition, the number and function of peripheral Tregs decrease, indicating that suppression of the T cell response is still clinically crucial at the stage of full progression of the disease [32].

The effect of NS on T cells in patients with Hashimoto’s thyroiditis has not been studied so far. However, Farhangi et al. [24] showed that supplementation with powdered NS significantly reduces serum TSH and TPO-Ab concentrations in HT patients. Therefore, we decided to analyze the influence of NSEO on some essential properties of T cells from HT patients. We focused on examining T cells’ proliferation capacity, susceptibility to apoptosis and necrosis, and ability to produce cytokines. To do that, we prepared an experimental model in which human PBMCs from healthy women and HT patients were preactivated by 24 h stimulation with an immobilized monoclonal anti-CD3 antibody. Next, different NSEO dilution variants were added to the cell culture. The purpose of such an experimental protocol was to mimic in vivo conditions in HT patients, where lymphocytes are continuously activated upon contact with an autoantigen. The incredible advantage of this model was the use of PBMCs, which are a mixture of different cells; in addition to T cells, there are also B cells, dendritic cells (DCs), and monocytes. In HT, the role of not only T cells but also of other cells that participate in the presentation of autogens and stimulate antibody production is essential.

Our findings showed that the proliferation of CD4^+^ and CD8^+^ T cells from healthy women and HT patients is significantly inhibited mainly in the presence of the lowest (1:10) NSEO dilution; the number of cell divisions reduced, and the percentage of dividing cells decreased. In our previous work, we used a different stimulation protocol in which NSEO was added simultaneously with anti-CD3 antibody, which resulted in a much more significant reduction in the percentage of proliferating T cells from healthy people, observed even at higher NSEO dilutions (1:50 and 1:100) [29]. Thus, adding NSEO before stimulation has a much stronger effect, probably due to the inhibition of T cell activation through the TCR/CD3 complex. Meanwhile, the effect of NSEO on previously activated T cells is weaker due to earlier activation of specific signaling pathways essential for their function.

Our results would explain the observation made by Kheirouri et al. [19], who demonstrated that oral supplementation with NS oil decreases the percentage of blood CD8^+^ T cells in RA patients. It could also explain a reduction in RA symptoms after oral NS supplementation. Similar observations were made in other immune-related diseases; symptoms reduced after consuming NS seeds or oil in patients with asthma [33] and allergic rhinitis [18,34]. The common feature of these diseases is the malfunction of CD4^+^ T cells, which react to antigens that usually do not cause any reaction in healthy people (allergens) or autoantigens present in the body and then induce a response of CD8^+^ T cells and B cells.

A reduction in the percentage of proliferating T cells in the presence of 1:10 NSEO dilution in cell culture was related to intensified cell apoptosis and necrosis. Cell death was also induced in the presence of 1:50 NSEO. Similar effects were seen in HT patients and healthy women. Similar cytotoxic activity was also observed for methanolic NS seed extract in human lymphoma U937 cells [35], SiHa human cervical cancer cells [36], and MCF- 7 breast cancer cells [37]. NS inhibited cell proliferation and stimulated apoptosis by activating p53 and caspases. However, there are some papers describing the protective effect of NS. For instance, Tripathi and Pandey [38] showed that the methanolic extract from NS seeds protects human lymphocytes stimulated with phorbol myristate acetate (PMA) from apoptosis. Salem et al. [39] showed that NS oil increases the numbers of CD4^+^ T cells but not CD8^+^ T cells in mice.

It has been suggested that thymoquinone could be the primary bioactive constituent of NSEO as it induces mitochondria-mediated apoptosis of leukemic cell lines through the Bax pathway [40,41]. Another proposed active compound is α-hederin, which causes the formation of membrane pores and changes in the cell membrane, including vacuolization of the cytoplasm, leading to cell death in cultures of cancerous (melanoma) and non-cancerous mouse 3T fibroblast cells [42,43]. However, essential oils of plants are a mixture of volatile substances of different natures and complex chemical compositions. Their rich chemical composition determines their multidirectional biological activity. As substances with high lipophilicity, they easily penetrate the cell wall and membranes of various microorganisms, disrupting the integrity of these structures. One of the proposed mechanisms of toxicity against cells of EOs or their terpenoid components is the coagulation of the cytoplasm and, above all, the permeabilization of the cell membrane, which causes excessive loss of ions, which consequently lowers the membrane potential, disrupting the functioning of proton pumps and the associated reduction in the pool of intracellular ATP [44].

In Hashimoto’s thyroiditis, the immune response of Th1 lymphocytes predominates, which, together with macrophages, produce mainly pro-inflammatory cytokines, including INF-γ, TNF-α, IL-1, and IL-6. This mechanism activates the apoptosis of thyroid cells through FasL co-expressed on the cell surface [45]. In our study, we observed that cells from HT patients produced high levels of IL-10, TNF, IFN-γ, and IL-17A after stimulation with anti-CD3 antibody alone. Most of the listed cytokines are involved in developing Th1 and Th17 responses. While the immunopathogenesis of HT is mainly associated with excessive activity of Th1 cells and CTLs, it has been shown that HT patients have increased levels of circulating Th17 cells and that Th17 cells are found in the thyroid gland [46,47]. IL-17, a pro-inflammatory cytokine, induces the production of other pro-inflammatory cytokines, such as IL-1β and IL-6 [48]. An increase in the same cytokines was also observed in healthy women in this study, indicating a strong activation of Th1 and Th17 responses under culture conditions.

Our results also demonstrated that Th1, Th2, and Th17 cells do not produce cytokine synchronously, regardless of health status. The dynamics of cytokine secretion vary [49], which explains why IL-10, TNF, IL-6, IFN-γ, and IL-17A increased in cell culture supernatants after 72 h of stimulation with anti-CD3 antibody alone, while IL-2 and IL-4 were low. The secretion of IL-2 is a critical and early event in the activation program of T cells. The production of this critical cytokine is subject to multifactorial regulation at the level of chromatin remodeling, transcription, message stability, and possibly even translation. In vivo studies using animal models have demonstrated that the transcript can be detected in naive T cells as early as 1 h after stimulation and peaks at 4–6 h [50]. However, maximal intracellular accumulation of the protein in vivo has been reported to take as long as 12–14 h [51]. Next, IL-2 production terminates quickly, and secretion is undetectable by 20–24 h in either cell type [52]. This explains why IL-2 in the supernatants was low after anti-CD3 antibody stimulation in HT patients and healthy women in this study. Studies show that the secretion of IL-4 is generally low compared to IFN or IL-17 [49], which we also saw in our model.

Noteworthy was the significant decrease in IL-17A production in the presence of the broad spectrum of NSEO dilutions in HT patients and healthy women. It was recently shown in a mouse model that B cells induce the activation of Th17 cells [53], and these cells are inhibited by brodalumab in the treatment of various autoimmune diseases, such as systemic sclerosis [54]. In our study model, among PBMCs, B cells were found. From our observations, B cell proliferation is also inhibited in the presence of NSEO. Therefore, they probably contribute to IL-17A secretion in both healthy women and HT patients.

In healthy women, the presence of 1:10 and 1:50 NSEO dilutions in cell culture stimulated the release of IL-2 and IL-4 into cell culture supernatants. In HT patients, the concentration of IL-4 was higher in the presence of 1:50 NSEO, without affecting the IL-2 levels. These results could suggest that NSEO shifts the T cell response toward Th2, as evidenced by the increased IL-4 levels. Barlianto et al. [55] obtained the opposite effect while examining the effect of NS oil supplementation in children with asthma. The authors demonstrated that 8 weeks of NS oil supplementation increases serum IFN-γ levels and reduces IL-4 levels, thus redirecting the T cell response toward Th1. The discrepancy in the results may be because HT is dominated by the Th1 response, which under the influence of NSEO is redirected toward Th2. Meanwhile, in asthma, the Th2 response is dominant, which under the influence of NS may have been redirected toward Th1, proving the immunomodulatory properties of NSEO. However, it should be emphasized that the authors analyzed serum cytokine levels after NS oil supplementation, while we assessed the effect of NSEO in vitro. Therefore, it seems more likely that the release of IL-2 and IL-4 into cell culture supernatants resulted from the cytotoxic effect of NSEO on lymphocytes in the tested dilutions.

Still, the phenomenon cannot be only explained by intensified cell death because some cytokines (e.g., IFN-γ, IL-6) were unaffected by NSEO and some (e.g., IL-17A) decreased in the broad spectrum of NSEO dilutions, even those that did not cause apoptosis. In addition, the sensitivity of cells to NSEO was also dependent on health status. For example, TNF was not significantly affected by NSEO in healthy women, while it decreased in the presence of 1:10 and 1:100 NSEO dilutions in HT patients. In contrast, IL-10 decreased in the presence of 1:10–1:100 NSEO dilutions in healthy women, but in HT patients, it remained unaffected.

## 5. Conclusions

Even though the size of the study groups was relatively small, we demonstrated that essential oil sourced from *Nigella sativa* seeds has a proapoptotic and antiproliferative effect on activated T cells in patients with Hashimoto’s thyroiditis under cell culture conditions. In addition, NSEO also affects the Th1/Th2/Th17 cytokine production by reducing the concentration of some cytokines important for regulating inflammatory processes, such as TNF and IL-10, and the Th1 response, i.e., IFN-γ, and increasing the production of IL-4 associated with the Th2 response in healthy women and HT patients.

It is unquestionably true that the differences in the T cell reaction between HT patients and healthy participants are relatively small. The exceptions are changes in the concentrations of IL-2 and IL-4 in cell culture supernatants in response to NSEO. However, we believe that even those modest differences may reflect a changing trend and potentially affect Hashimoto’s thyroiditis. Furthermore, HT patients are predisposed to other autoimmune conditions, which means potential autoreactive cells are present among blood lymphocytes. Further research with a broader range of measures and a larger study group is needed to fully understand the significance of these differences between the study and control groups.

To the best of our knowledge, this is the first scientific study that explains the mechanisms of action of *Nigella sativa* on immune cells. Furthermore, it shows that NSEO has a strong immunomodulatory effect that could favorably affect the abnormal response of T cells in immune-related diseases, such as allergies or autoimmune syndromes. Still, additional studies are necessary to explain how exactly NSEO influences lymphocytes and what signaling pathways it affects.

## Figures and Tables

**Figure 1 antioxidants-12-01246-f001:**
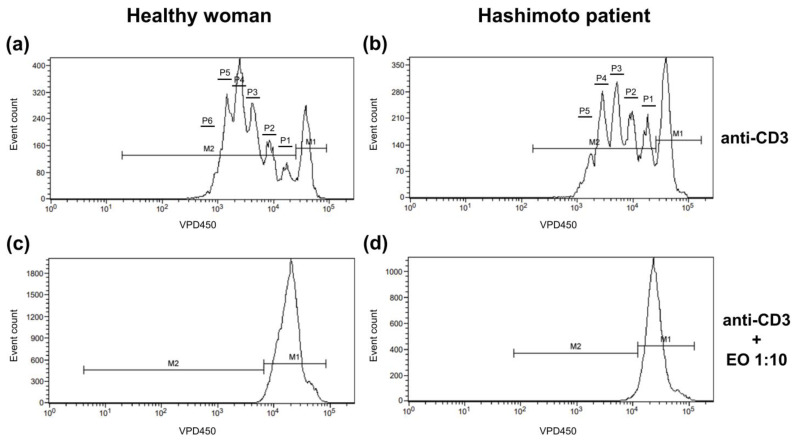
Representative histograms showing the proliferation pattern of CD4^+^ cells stimulated with anti-CD3 antibody alone (**a**,**b**) or with 1:10 NSEO (**c**,**d**). Graphs show cell proliferation measured as VPD450 fluorescence. Marker 1 (M1) indicates non-diving cells; M2, the proportion of cells proliferating (daughter cells) in response to stimulation; and P1–P6, subsequent cell divisions.

**Figure 2 antioxidants-12-01246-f002:**
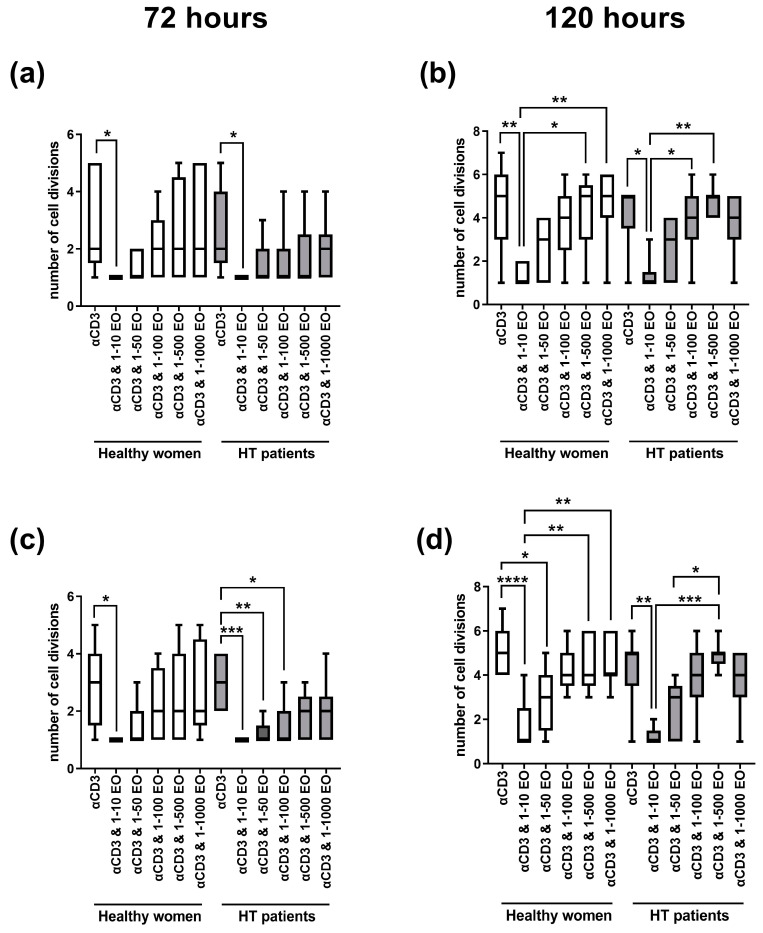
Comparison of the number of cell divisions of CD4^+^ (**a**,**b**) and CD8^+^ (**c**,**d**) cells stimulated with anti-CD3 antibody alone (control) or with different dilutions of NSEO for 72 (**a**,**c**) and 120 (**b**,**d**) hours in healthy people and HT patients. Graphs show the median, percentiles with the maximum and minimum values, and ANOVA Friedmann with Dunn’s post hoc test; * *p* < 0.05, ** *p* < 0.01, *** *p* <  0.001, **** *p* < 0.0001.

**Figure 3 antioxidants-12-01246-f003:**
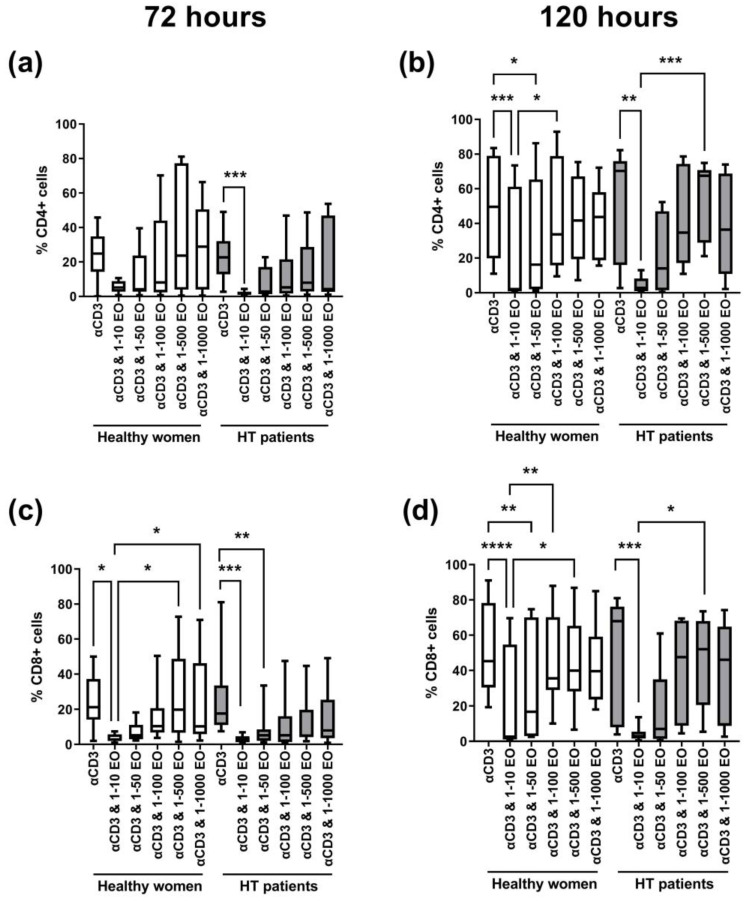
Comparison of the percentage of proliferating CD4^+^ (**a**,**b**) and CD8^+^ (**c**,**d**) cells stimulated with anti-CD3 antibody alone (control) or with different dilutions of NSEO for 72 (**a**,**c**) and 120 (**b**,**d**) hours in healthy people and HT patients. Graphs show the median, percentiles with the maximum and minimum values, and ANOVA Friedmann with Dunn’s post hoc test; * *p* < 0.05, ** *p* < 0.01, *** *p* < 0.001, **** *p* < 0.0001.

**Figure 4 antioxidants-12-01246-f004:**
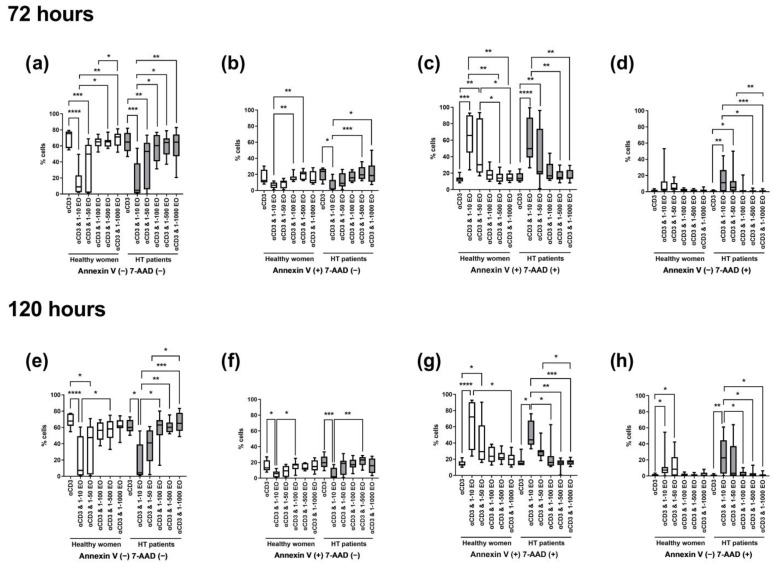
Comparison of the percentage of living, apoptotic (early and late), and necrotic lymphocytes stimulated for 72 (**a**–**d**) and 120 (**e**–**h**) hours with anti-CD3 antibody (control) with different dilutions of NSEO in healthy people and HT patients. Graphs show the median, percentiles with the maximum and minimum values, and ANOVA Friedman with Dunn’s post hoc test; * *p* < 0.05, ** *p* < 0.01, *** *p* < 0.001, **** *p* < 0.0001.

**Figure 5 antioxidants-12-01246-f005:**
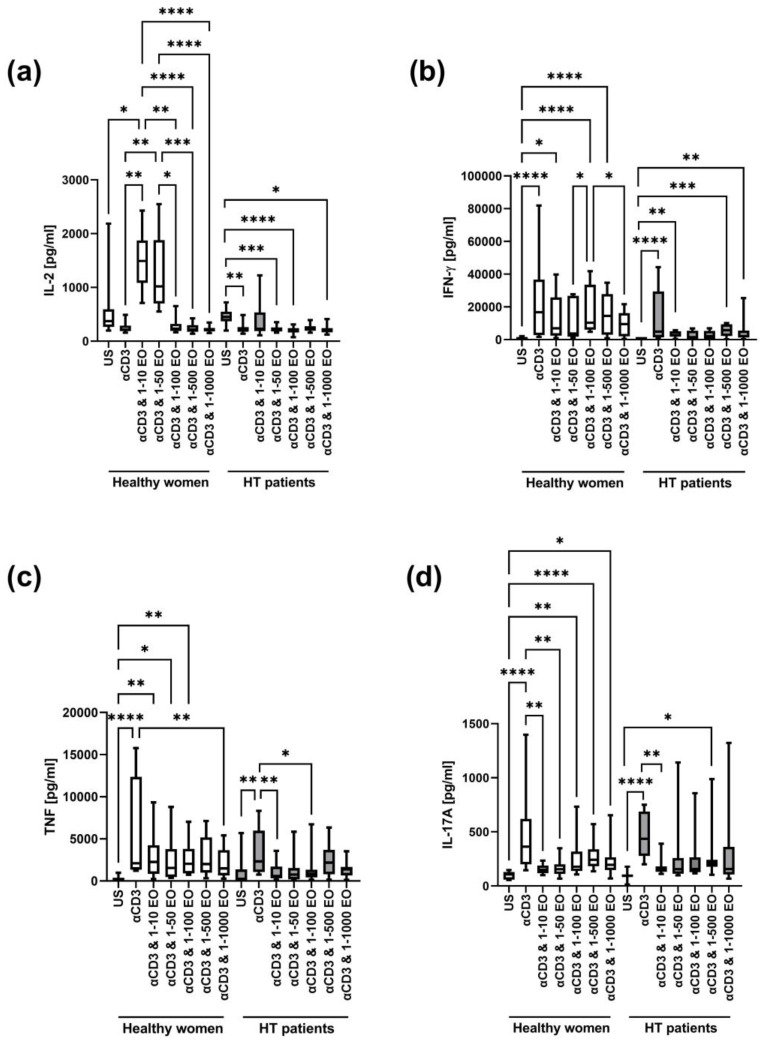
Comparison of Th1 and Th17 cytokine levels produced by cells stimulated for 72 h with anti-CD3 antibody with different dilutions of NSEO in healthy controls and HT patients. Graphs show the levels of IL-2 (**a**), INF-γ (**b**), TNF (**c**), and IL-17A (**d**). Graphs show the median, percentiles with the maximum and minimum values, and ANOVA Friedmann with Dunn’s post hoc test; * *p* < 0.05, ** *p* < 0.01, *** *p* < 0.001, **** *p* < 0.0001. US—unstimulated.

**Figure 6 antioxidants-12-01246-f006:**
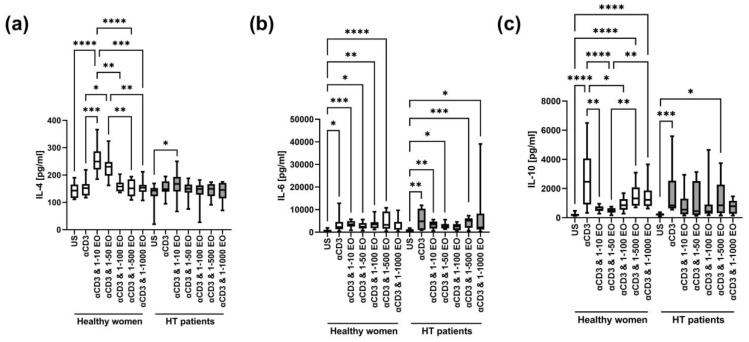
Comparison of Th2 cytokine levels produced by cells stimulated for 72 h with anti-CD3 antibody with different dilutions of NSEO in healthy controls and HT patients. Graphs show the levels of IL-4 (**a**), IL-6 (**b**), and IL-10 (**c**). Graphs show the median, percentiles with the maximum and minimum values, and ANOVA Friedmann with Dunn’s post hoc test; * *p* < 0.05, ** *p* < 0.01, *** *p* < 0.001, **** *p* < 0.0001. US—unstimulated.

## Data Availability

The data presented in this study are available upon request from the corresponding author.

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
