# Peer review of "The Impact of Nigella sativa Essential Oil on T Cells in Women with Hashimoto’s Thyroiditis"

_antioxidants, 2023, doi:10.3390/antiox12061246_

Round 1

Reviewer 1 Report (Previous Reviewer 2)

The authors have adequately answered the questions. I have nothing to add in particular.

Reviewer 2 Report (Previous Reviewer 1)

The manuscript has been improved

This manuscript is a resubmission of an earlier submission. The following is a list of the peer review reports and author responses from that submission.

Round 1

Reviewer 1 Report

The manuscript by Ciesielska-Figlon et al is focused on an interesting and practically significant issue of the effects of natural compounds (supplements) on the immune response in Hashimoto's disease. The study is very thorough and systematic. The data are mostly convincing. The report is written clearly and logically.

There are a few issues, however, that need to be addressed to improve this report.

General issues:

1. Is suppression of T-cell responses still clinically relevant at the stage of fully developed Hashimoto's thyroiditis? What is the degree of thyroid's destruction at this point?

2. PBMC, not purified T cells, have been used in the experiments. This is quite justified considering that T cells response in vivo interacting with other cells. However, an acknowledgement of this fact is required, and the discussion has to be adjusted accordingly. It is very likely that at least some effects of NS oil in this study are attributed to cells other than T lymphocytes.

3. The differences between patients with HT and healthy participants determined in this study appear to be very modest. This fact does not invalidate the significance of findings, since a general effect on healthy T cells may well be beneficial in HT. However, this finding needs to be commented on and discussed.

Specific points:

4. What is the concentration of NS oil in the samples? Only dilutions are shown. What is the number of cells per well/sample? What is a sample volume? 

5. Fig. 1, 2: What is the definition of proliferating cells and of the number of proliferations? In addition to having clear definitions for both parameters, it would be very helpful to see the corresponding flow data (at least for some points) for illustration.

6. Fig. 3: More than 50% of T cells show advance apoptosis or necrosis at a 1:10 dilution of NS oil. This indicates a high toxicity level of this preparation. was it in any way related to the effect of ethanol? Was ethanol added to the control sample? Generally, this fact needs to be acknowledged and commented upon. A related question is as follows: what is a typical concentration of NS oil in blood after oral administration of either oil or seeds?

7. Fig. 4: NS oil appears to suppress cytokine production by PBMC. But IL-2 is an exception from this rule. Furthermore, anti-CD3 stimulation does not induce IL-2 production. This is unusual and deserves a special discussion. Why anti-CD3 does not stimulate IL-2 production by PBMC and why NS oil at high concentrations facilitate this production in healthy women?

8. The same seems to be true for IL-4 (Fig. 5). Why?

9. The following few questions are related to the Th1/Th2 classification of the responses studied in this work. In Fig. 5 legend IL-6 is named a Th2 cytokine. IL-6 is not a Th2 cytokine, and generally is not a T-cell-produced cytokine. There is nothing strange about non-T-cell cytokines being affected when PBMC are examined (see #2).

10. Line 339: IL-12 is not produced by Th1 cells. It promotes differentiation toward Th1, but is not produced by them.

11. Line 400: An increase in IL-2 and IL-4 is considered indicative of a shift toward Th2, but IL-2 is clearly a Th1 cytokine. Furthermore, the dilutions generating this effect are linked to considerable toxicity (see #6). Do we really see a Th1->Th2 shift?

Reviewer 2 Report

In this article, Figlon et al examined the effect of essential oil from Nigella sativa (NSEO) on T cells from HT patients, especially their proliferation capacity, ability to produce cytokines, and susceptibility to apoptosis. The lowest ethanol (EtOH) dilution (1:10) of NSEO significantly inhibited the proliferation of CD4+ and CD8+ T cells from HT patients and healthy women by affecting the percentage of dividing cells and the number of cell divisions. In addition, 1:10 and 1:50 NSEO dilutions induced cell death. Different dilutions of NSEO also reduced the concentration of IL-17A, IL-10, TNF, and IFN-γ in cell culture supernatants from HT patients and healthy women. Almost all NSEO dilutions increased the concentration of IL-6 in both study groups. In healthy women, the level of IL-4 and IL-2 was significantly increased in the presence of 1:10 and 1:50 NSEO dilutions. They concluded that NSEO has a strong immunomodulatory effect on the lymphocytes of HT patients. There have been few studies on Nigella sativa essential oil on T cells in women with Hashimoto's thyroiditis, and the results are very valuable. I have some questions about what I was reading that I didn't understand.

major concerns)

1) This is a very nice study that shows the differences in NSEO in t-cells, especially in comparison between healthy individuals and patients with autoimmune diseases. Although the detailed mechanisms by which these phenomena occur remain unclear, it is clear that the effects on cell death, including cytokine production and apoptosis/necrosis, are particularly pronounced at high doses of NSEO. 

 In relation to the above, how significant are these concentrations in vivo? If they are to be used in a therapeutic application (oral or topical?), are they realistic concentrations in either a systemic or local sense?

2) I understood the phenomenology, but not the detailed mechanism of action of NSEO. I would like to see some discussion on these, even hypothetical. Is there anything that has been said about it? Unlike Nigella damascena, NSEO seems to be known to contain plant alkaloids with low toxicity, please provide a discussion of the components and mechanism of action considered to date.

3) In the Discussion section, the relationship between cytokines such as IL-17 and HT, collagen disease, and T and B cells is discussed. Recently, it has been reported that B cells induce Th17 cells in the interaction between B cells and T cells (Single-cell-level protein analysis revealing the roles of autoantigen-reactive B lymphocytes in autoimmune disease and the murine model. Elife. 2021 Dec 2;10:e67209. doi: 10.7554/eLife.67209. PMID: 34854378; PMCID: PMC8639144.). In other collagen diseases, brodalumab, an anti-IL-17RA receptor antibody, has been reported to improve the pathogenesis of SSc, including skin sclerosis, immune abnormalities, and vascular disorders in systemic sclerosis (Interleukin-17 pathway inhibition with brodalumab in early systemic sclerosis: analysis of a single-arm, open-label, phase 1 trial. Journal of the American Academy of Dermatology. 2023.). Citing them, please discuss the importance of B cells and T cells in collagen diseases such as HT.

minor concerns)

1) In line 554-556, you wrote "Disclaimer/Publisher’s Note: The statements, opinions and data contained in all publications are solely those of the individual author(s) and contributor(s) and not of MDPI and/or the editor(s). MDPI and/or the editor(s) disclaim responsibility for any injury to people or property resulting from any ideas, methods, instructions or products referred to in the content." However, I wonder if this sentence should be brought before REFERENCES section. Please correct accordingly.